# Surface Modification on Polyimide Yarn by Plasma Treatment to Enhance Adhesion with Polypropylene Resin

**DOI:** 10.3390/polym14194232

**Published:** 2022-10-09

**Authors:** Hong Cui, Xiuli Gao

**Affiliations:** 1College of Textiles and Clothing, Yancheng Institute of Technology, Yancheng 224051, China; 2College of Textiles, Henan Institute of Engineering, Zhengzhou 450007, China

**Keywords:** polyimide yarn, surface modification, plasma, adhesion

## Abstract

Polyimide yarn as a kind of high performance fiber material has to improve the adhesion between the material surface and the resin in order to get a deeper application. The surface of polyimide yarn is modified by low temperature plasma treatment, and the effect of plasma treatment parameters on the adhesion between polyimide yarn and polypropylene resin is studied. By comparing the extraction force on the surface of polyimide yarn before and after treatment, the effect of plasma treatment parameters such as treatment time, processing gas and treating power on yarn adhesion is investigated. Furthermore, the adhesive force between polyimide yarn and polypropylene resin is analyzed by a single factor to optimize the process parameters to obtain higher adhesive force. Additionally, the Box–Behnken design is utilized to optimize the plasma treatment parameters, and the significance of the influence of the plasma treatment parameters on the adhesion between the polyimide fiber and the resin is discussed. The optimal process parameters are obtained through analysis: the treatment time 90 s, the processing gas oxygen, and the treating power 150 W.

## 1. Introduction

Firstly, plasma can be applied to fiber surface modification. Plasma modification effect on fiber surfaces is closely related to the plasma treatment voltage [1]. The influence of plasma voltage on the treatment effect is studied. Helium plasma is used to treat ramie fiber, and the influence on the modification of ramie fiber is studied by changing plasma voltage. Atmospheric dielectric barrier discharge plasma is used to modify the ethanol pretreated ramie fiber to improve the interface adhesion between hydrophilic ramie fiber and hydrophobic polypropylene matrix [2]. The results are obtained that ethanol flow rate is the most important parameter affecting the plasma modification effect of ramie fiber.

In addition to ramie, fiber surface modification can also be applied to other fibers such as short glass fiber, carbon fiber, cellulose fiber, polyimide fiber, and aramid fiber by air plasma. Short glass fibers are characterized by low surface energy and poor wettability [3]. Wet glass fiber reinforced epoxy resin and polyester composites can be obtained by plasma treatment of glass staple fibers, and the bonding properties between them can be improved. It is shown as the polarity of the fibers can be improved by plasma modification and the morphology of the fibers becomes rough. Carbon fiber is modified by 30 Pa and 200 W air plasma at different treatment times [4]. The effect of low pressure plasma treatment on cellulose fibers [5] is studied in order to improve the adhesion between polymer matrix and natural fibers. The surface of cellulose fibers is successfully modified using the air plasma treatment to improve the adhesion of thermoplastic starch matrix and fiber [6]. The results indicate a significant improvement of adhesion between treated cellulose fibers and thermoplastic starch matrix by tensile and scanning electron microscopy images of the fracture surfaces.

The surface modification of the above fiber is carried out by air plasma, while the polyimide fiber and aramid fiber, including date palm fiber, are treated by oxygen plasma. The surface behavior of polyimide fiber reinforced epoxy composite is prepared by using oxygen plasma. The results indicate that the surface roughness of polyimide fiber is elevated by the introduction of oxygen functional groups [7]. Moreover, the surface free energy of the fibers and the interfacial adhesion of the composites are greatly improved after oxygen plasma treatment, and the surface free energy of the fibers and the interfacial adhesion of the composites are greatly improved. Aramid fiber is modified by oxygen plasma treatment [8]. The effects of oxygen plasma treatment power on fiber surface and domestic aramid fiber reinforced bismaleimides composite interfacial properties are investigated, respectively. Due to the oxidation reaction and plasma etching after oxygen plasma treatment, new oxygen-containing groups C = O and -COO are introduced, which improve the surface roughness of fiber, change the surface morphology of fiber, and improve the wettability of fiber surface effectively. Oxygen plasma with different plasma discharge power and exposure time [9] is used to treat date palm fiber. As a result, the plasma treatment can clean the fiber surface and improve the surface roughness through corrosion. In addition, the fiber surface modification significantly improved the tensile properties and the interfacial shear stress of date palm fiber.

Secondly, plasma can be applied to fabric surface modification. The effect of plasma surface modification is closely related to fabric construction and surface structure [10]. Polypropylene fabric with a polyurethane coating is treated by atmospheric plasma, and the process parameters are optimized to maximize the adhesion between nylon 66 and polyurethane [11]. The outcomes show that it is feasible to apply atmospheric pressure plasma as a new technology in the industrial textile field. Adhesion property of polyester fabric to a silicone rubber coating is studied by plasma treatment [12]. The results demonstrate that argon and air plasma can enhance the adhesion between the fabric and the coating. The surface properties of inkjet printing water-based polyester pigment ink are modified by atmospheric-pressure air/He plasma to improve the color strength and pigment adhesion of treated surfaces [13]. Compared with air plasma treatment, the polyester fabric treated by air/He hybrid plasma has better surface properties and inkjet printing performance.

Conductive fabrics are prepared by plasma treatment of polyester fabrics to enhance the adhesion of reduced graphene oxide [14]. The combination of polypyrrole with polyester film and fabric improved by low pressure oxygen plasma is systematically studied [15]. The effects of plasma on surface roughness, surface chemistry and hydrophilicity are studied by using different plasma treatment times. The results show that the increase of surface carboxyl functionalization and the formation of nanoscale roughness contribute to the improvement of adhesion and conductivity by polyester surface morphology.

Atmospheric plasma treatment is mainly studied to improve the mechanical properties of polylactic acid fiber bonding joints used in packaging industry in order to improve the mechanical properties of polylactic acid fiber bonding joints [16]. The effects of helium/oxygen atmospheric plasma pretreatment conditions, including treatment time, oxygen flow rate and the distance between nozzle and sample on sizing performance of cotton yarn are studied [17]. The results confirm that plasma treatment can effectively improve the surface roughness, static friction coefficient, wettability of raw cotton fiber and the adsorption capacity of cotton yarn to starch size. Therefore, the sizing adhesion and elongation at break of starch sizing roving are greatly affected by the treatment mode.

The influence of plasma polymerization process on the fiber/matrix interface strength is studied in order to improve the mechanical properties of basalt fiber/epoxy composite [18]. A multilayer film [19] is prepared by using atmospheric cold plasma as an adhesion improvement agent between polycaprolactone or polylactic acid and starch layers. The treatment of polycaprolactone and polylactic acid films increases the surface roughness and reduces the water contact angle. The bonding properties and non-bonding properties of polytetrafluoroethylene treated with or without heated plasma are compared with different plasma treatment times [20]. Adhesion-free adhesion requires a longer time of heat-assisted plasma treatment than that of adhesives in order to achieve high adhesion strength.

At present, polyimide fibers are mostly prepared by electrospinning and some practical applications have been obtained. However, in general, the density of electrospun fibers has a very low density, and the concept of “self-bonding” is used to prepare high-density polyimide fiber sponges, with the density up to 280 mg/cm^3^, porosity up to 80%, and significantly increased compressive strength [21]. The application of polyimide fiber is combined with more new materials and new technology application in composite materials. In terms of new materials, ternary polyimide electrospinning composite nanofibers are prepared by ternary polyamine acid composite precursor, and composite materials with a higher tensile strength and glass transition temperature are obtained, which can be used in the fields of filtration, fiber composite material, and battery [22]. In addition, the addition of phosphorus-containing compounds can be used as plasticizers to prepare electrospun polyimide nanofibers, which can obtain excellent mechanical and thermal properties and can be used to prepare lithium-ion battery separators, which have a higher puncture strength compared with other lithium-ion battery separators [23]. In terms of the combination with new technology, the vacuum-assisted hot pressing method is used to enhance the mechanical properties of polyimide nanofiber/epoxy composites, and the morphology of polyimide nanofiber/epoxy composites is observed. The analysis of thermal and mechanical properties show that the mechanical properties are significantly improved [24]. Furthermore, using the method of in-situ polymerization and in situ thermal conversion to the preparation of electrostatic spinning polyimide/reduction of graphene oxide composite nanofibers [25], the fiber has high tensile strength and modulus, and also has good thermal stability and glass transition temperature; for the preparation of high performance composite nanofibers, electrospun nanofibers provide a new path. Polyimide fiber can also be modified by plasma treatment to change the adhesion between polyimide fiber and resin, which provides a reference for the further application of polyimide fiber in the field of composite materials.

In this paper, the surface of polyimide yarn is modified by plasma, and the effect of plasma treatment parameters on the adhesion between polyimide yarn and polypropylene resin is studied. Through the response surface design of treatment conditions such as power, time, and gas source, the combined effects of process parameters and interactions on the adhesion between polyimide yarn and polypropylene resin are investigated. The plasma treatment parameters are optimized for the best adhesion performance. It not only solves the problems of poor wettability and adhesion of polyimide yarn as reinforcing materials, but also further expands the application field of polyimide yarn.

## 2. Materials and Methods

### 2.1. Material and Methodology

The materials used in the experiment are as follows: polyimide yarn, polypropylene yarn (used as thermoplastic resin), high temperature corrugated paper, acetone, and deionized water. The polyimide yarn is treated on an HD-1B cold plasma modification device (Changzhou Zhongke Changtai Plasma Technology Co., Ltd., Changzhou, China). First, the polyimide yarn is washed with acetone for 30 min to remove impurities such as surface oil, and then dried at environment temperature for 6 h to fully volatilize the acetone. Then, use the polypropylene yarn (used as thermoplastic resin) to knot the polyimide yarn (the sample being made is shown in the Figure 1), and then put it into the HD-1B cold plasma modification device for processing. The type of plasma used is low temperature plasma. The extraction force between the treated polyimide yarn and polypropylene resin is tested with XQ2 fiber strength elongator (Donghua University) to characterize adhesion between the polyimide yarn and polypropylene resins. The extraction force test device is as shown in Figure 2. The greater the extraction force, the stronger the cohesion, and the smaller the extraction force, the worse the cohesion.

### 2.2. Effect of Plasma Treatment on Adhesion between Polyimide Yarn and Polypropylene Resin

By comparing the extraction force of polyimide yarn and polypropylene resin before and after plasma treatment, it is reflected whether plasma treatment has a significant effect on the extraction force of polyimide yarn and polypropylene resin.

### 2.3. Single Factor Study on the Adhesion between Plasma-Treated Polyimide Yarn and Polypropylene Resin

The other two process parameters among the three process parameters of plasma processing time, output power, and gas source were fixed, and one of the process parameters was changed. The influence of the change of process parameters after plasma treatment on the extraction force between polyimide yarn and polypropylene resin was analyzed.

### 2.4. Study on the Optimization Process of Plasma Treatment of Polyimide Yarn and Polypropylene Resin

Based on the results obtained in the previous stage, the Box–Behnken design (BBD) is chosen as the experimental design for the optimal process [14]. Box–Behnken is one of the most commonly used response surface optimization methods, which uses a reasonable experimental design method, obtains certain data through experiments, uses the functional relationship between the fitting factor and the response value of the multivariate quadratic regression equation, and seeks the optimal process parameters through regression equation analysis to solve a multivariate problem. It is a method for optimizing experimental conditions and is suitable for solving problems related to nonlinear data processing. Through regression fitting, drawing response surfaces, and contour lines, the response values corresponding to each factor level can be obtained. According to the response value of each factor level, the predicted optimal response value and the corresponding experimental conditions can be found. Table 1 is the factor level coding.

## 3. Results and Discussion

### 3.1. Effect of Plasma Treatment on the Adhesion between Polyimide Yarn and Polypropylene Resin

Process parameters such as treatment time (90 s), treatment power (100 W), and gas source (Oxygen) are selected to treat the polyimide yarns, and the extraction force of the untreated polyimide yarns is compared. The results are shown in Figure 3.

As shown in Figure 3, the extraction force between the polyimide yarn and the polypropylene resin after the treatment increased by 68% compared with that before the treatment, so the plasma treatment has a significant effect on the cohesion between the polyimide yarn and the polypropylene resin. Further single-factor experiments can be carried out to explore the effect of plasma treatment on the adhesion of polyimide yarn to polypropylene resins.

### 3.2. Effect of Single Factor on Cohesiveness between Polyimide Yarn and Polypropylene Resin

#### 3.2.1. Effect of Treatment Time on Cohesiveness

It can be seen from Figure 4 that, with the increase of the treatment time, the extraction force between the polyimide filament and the polypropylene resin first increases and then decreases. The reason is that, with the increase of treatment time, the polar groups on the surface of polyimide yarn increase, and the increase of active particles will lead to the increase of yarn surface pulling force, but after a certain treatment time, the deactivation of active particles will lead to a certain degree of decline in the extraction force. Therefore, the optimal treatment time is 120 s.

#### 3.2.2. Effect of Treatment Power and Gas on Cohesiveness

It can be seen from Figure 5 that, with the increase of the treatment power, the extraction force between the polyimide yarn and the polypropylene resin first increases and then decreases, and the optimal treatment power is about 200 W. This is because, with the increase of the discharge power, the active particles in the system increase, the energy obtained by the molecules increases, and the probability and strength of the active particles on the polyimide yarn surface are enhanced, so that more polar groups are combined to the surface of polyimide yarn, and the surface modification effect is increased. However, when the discharge power is too large, the average energy of active particles increases, and the collision between active particles leads to the quenching and inactivation of active particles, so that the extraction force decreases instead.

As can be shown from Figure 6, argon plasma has a higher extraction force between the polyimide yarn and polypropylene resin than oxygen plasma, which can effectively improve the surface cohesiveness of the polyimide yarn. Oxygen is a reactive gas. Oxygen on the surface of plasma can produce oxidation reaction on the material surface to bombard oxygen electrons into oxygen ions and free radicals, which can improve the infiltration and adhesion of the material surface. Argon is a non-reactive gas, which can easily form metastable atoms on the surface of the material during plasma treatment. The physical effect of high-energy particles can improve the polarity, infiltration, and cohesiveness of the material surface.

### 3.3. Optimum Treating Process of the Adhesion between Polyimide Yarn and Polypropylene Resin by Plasma

The optimized treatment time, power, and source gas in the previous stage were taken as factor levels to design the Box–Behnken experiment. The experimental design and results are shown in Table 2. ANOVA results of responses for plasma treated polyimide yarn are shown in Table 3.

There is still an obvious gap among the selected superior processes shown in Table 3. F value represents the significant degree of the influence of factors on the index. The larger F is, the more significant the influence of factors on the index is. In addition, *p*-value is an index to measure the difference. A *p*-value less than 0.05 indicates that factors have a significant impact on the index. The F and *p*-values of the extraction force and strain are analyzed, respectively. For the extraction force, the Model F-value of 3.3 implies that there is a 6.47% chance that an F-value could occur due to noise. *p*-values lower than 0.0500 indicate model terms are significant. In this case AB (Time-Power), A^2^ (Time^2^) are significant model terms. Values greater than 0.1000 indicate that the model terms are not significant. The results imply that the interaction between treatment time and power has the greatest effect on the extraction force. The effect of treatment time and power alone on the extraction force is not significant.

For the strain, the Model F-value of 2.0 implies that the model is not significant relative to the noise. There is a 18.67% chance that an F-value could occur due to noise. *p*-values less than 0.0500 indicate that model terms are significant. In this case A, C are significant model terms. Values greater than 0.1000 indicate that the model terms are not significant. The results mean that the treatment time and gas source have a significant influence on the extraction elongation, among which the treatment time has the greatest influence on the extraction elongation, the gas source has greater influence on the extraction elongation, and the power has the least influence on the extraction elongation.

Based on the developed model, three-dimensional response surface plots are generated using design expert 12.0 Stat-Ease. These plots showed the interactive effect of variables on the response. Experiments are carried out as per the design matrix (Table 2). Plots are generated by taking into consideration the effect of two factors at a time.

The extraction force between polyimide yarn and polypropylene resin shown in Figure 7a increases with an increase in power ranging from 100 W to 150 W, and extraction force also increases in plasma treatment time from 60 s to 90 s. Furthermore, it can be seen from Figure 7a–c that the extraction force between polyimide yarn and polypropylene resin increases first and then decreases in plasma treatment time from 60 s to 90 s, and the extraction force after argon treatment is less than that after oxygen treatment. Moreover, the extraction force between polyimide yarn and polypropylene resin decreases with an increase in power ranging from 100 W to 150 W, and the extraction force after argon treatment is less than that after oxygen treatment. It can be seen from the slope of the surface that the interaction between power and time has the greatest influence on the extraction force from Figure 7a–c, the interaction between gas source and time has the less influence on the extraction force, and the interaction between gas source and power has the minimal influence on the extraction force. Based on the above analysis, the optimal processing parameters of the plasma obtained are the treatment time being 90 s, the treatment power of 150 W, and the treatment gas being oxygen.

### 3.4. Verification of the Model

In order to verify the validity of the model, the obtained optimal plasma treatment parameters were tested on the machine, and the extraction force value was 126.6 cN. Compared with the extraction force value of 125.0 cN obtained by the prediction model, the difference rate was less than 5%, which has very high prediction accuracy.

## 4. Conclusions

In this paper, through a single factor test, the influence of each factor on the pulling force between polyimide yarn and polypropylene resin is investigated. The optimal process parameters of each factor are selected, and the optimization of process parameters is obtained by combining Box–Behnken experiment design. Finally, the configuration of process parameters with the best pulling force is obtained. After analysis, the following conclusions are drawn:

(1)In the single factor investigation stage, with the increase of plasma treatment time and power output, the adhesion between polyimide yarn and polypropylene resin increases continuously, and the increasing trend decreases gradually. This is because, with the increase of treatment time and power, the probability and strength of active particles on the surface of polyimide fiber yarn increase, so that the surface of polyimide yarn can bind more polar groups. However, when the treatment time and power continue to increase, the collision between active particles will lead to the inactivation and quenching of active particles, thus slowing down the surface modification effect.(2)In the process parameter optimization stage, through ANOVA and graphic analysis, the optimal plasma treatment process is as follows: reaction time is 90 s, treating power is 150 W, and reaction gas is oxygen. The optimized process parameters of plasma treatment show that oxygen plasma is a reactive gas under appropriate treatment time and power, which can produce a large number of free radicals and oxygen-containing groups on the surface of polyimide yarn during plasma treatment, which promote the modification effect of the polyimide yarn surface.

From the above conclusions, it can be seen that the low temperature plasma treatment technology has a significant surface modification effect on polyimide yarn, which can effectively improve the adhesion between polyimide yarn and polypropylene resin, so that polyimide yarn has a wider application space in aviation, aerospace, and other fields. At the same time, the plasma processing environment is friendly, easy to operate, and has good development prospects. This study can provide a reference for the further development and application of modified polyimide fiber yarns, so as to promote the further development of high-performance fiber resin matrix composites.

## Figures and Tables

**Figure 1 polymers-14-04232-f001:**
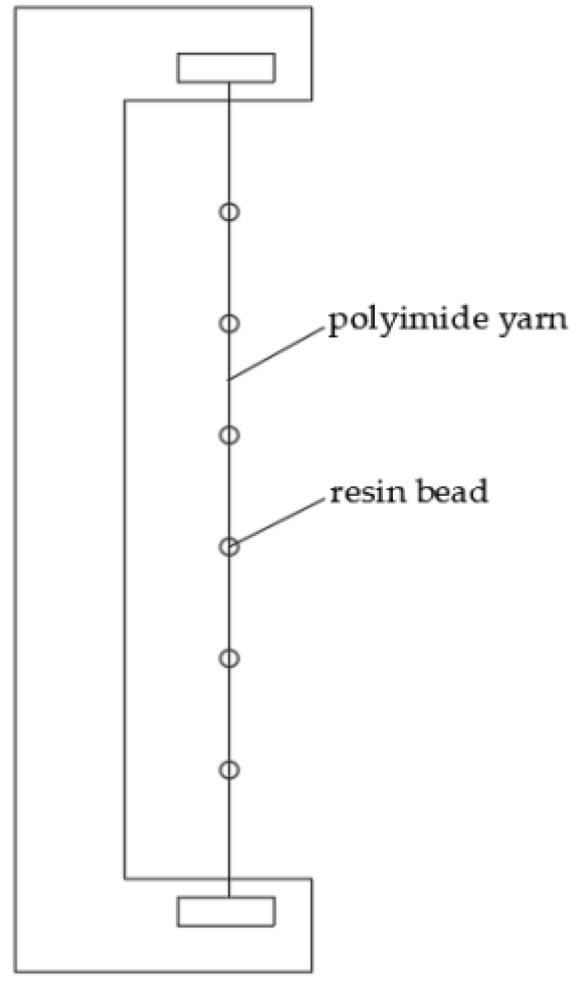
The sample being extracted.

**Figure 2 polymers-14-04232-f002:**
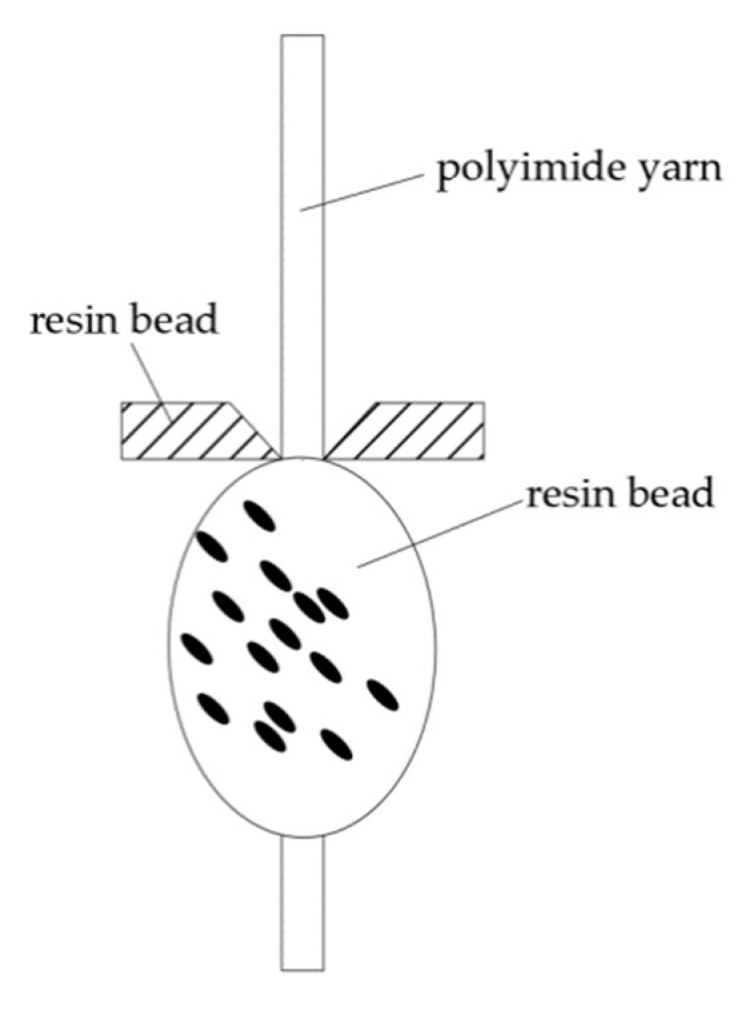
The extraction force test device.

**Figure 3 polymers-14-04232-f003:**
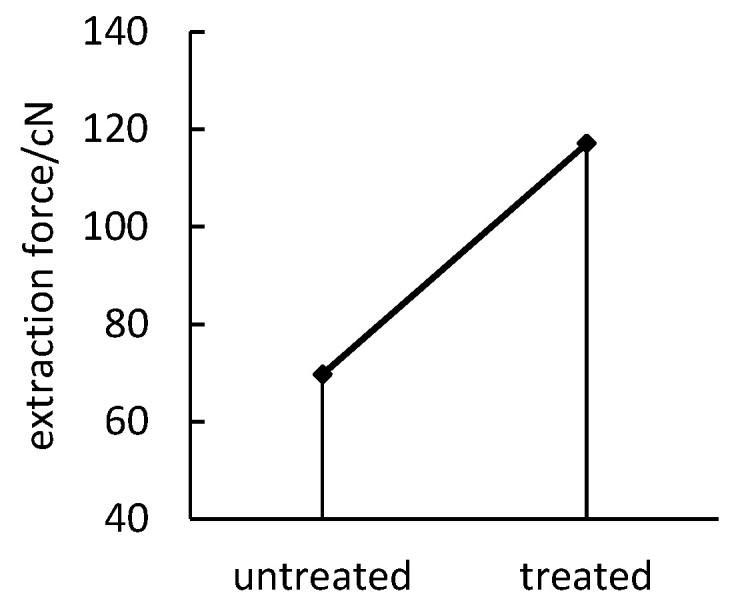
Comparison of extraction force before and after plasma treatment.

**Figure 4 polymers-14-04232-f004:**
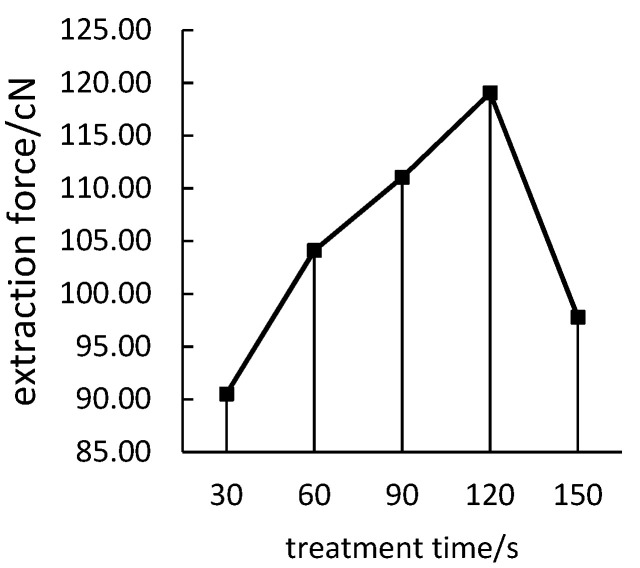
Effect of treatment time on the extraction force.

**Figure 5 polymers-14-04232-f005:**
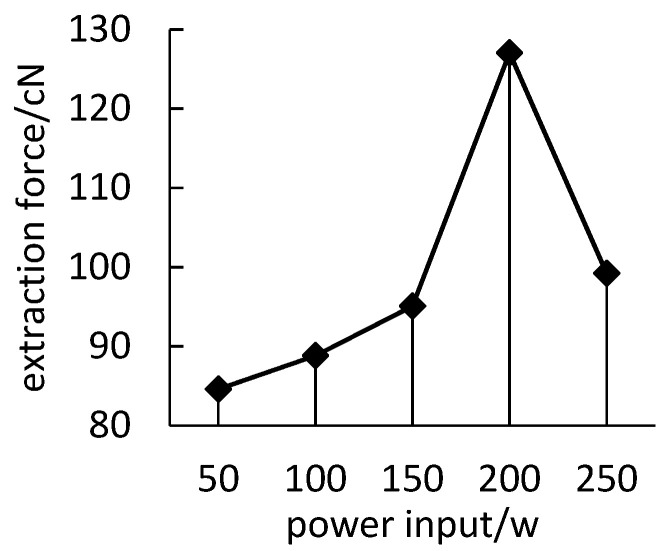
Effect of input power.

**Figure 6 polymers-14-04232-f006:**
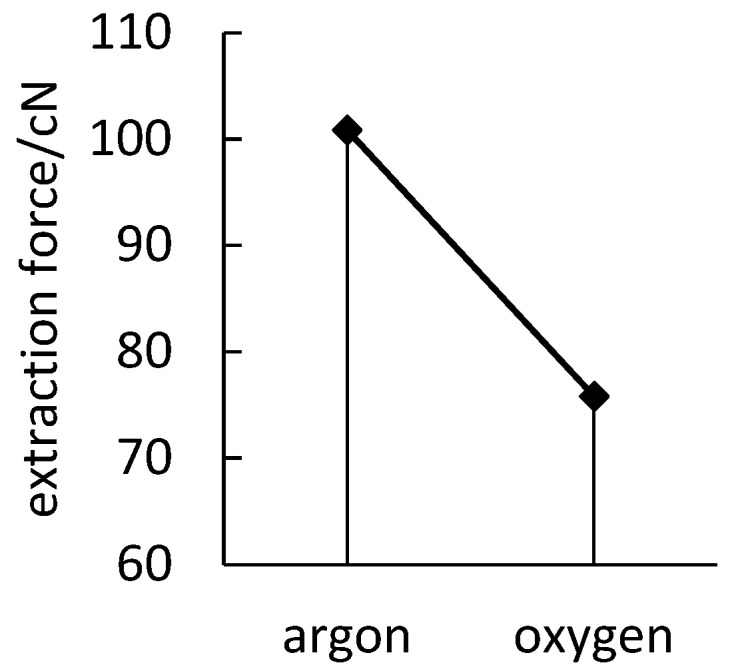
Effect of source gas.

**Figure 7 polymers-14-04232-f007:**
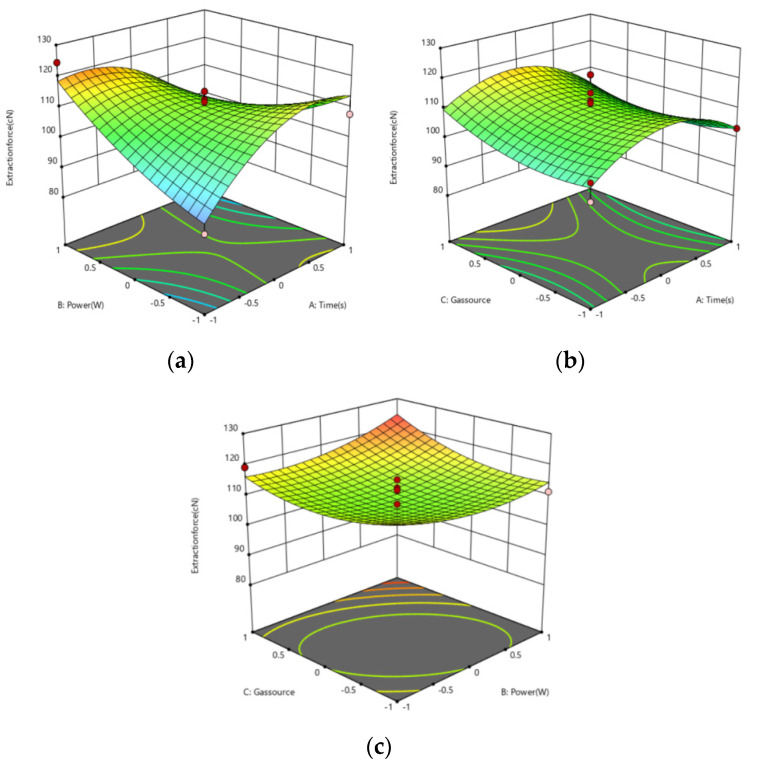
Response surface plot: Effect on extraction force of (**a**) power and time (**b**) gas and time (**c**) gas and power.

**Table 1 polymers-14-04232-t001:** Factor level coding.

Factor Code	Factor	Level Code		
		−1	0	1
A	Time/s	60	90	120
B	Power/W	100	150	200
C	Source gas	Argon	Oxygen	Argon

**Table 2 polymers-14-04232-t002:** Experimental design and results.

No.	Factor Level Coding			Actual Factor Level			Extraction Force/cN	Strain/%
	A	B	C	Time/S	Power/W	Source Gas		
1	−1	−1	0	60	100	Oxygen	88.39	5.5
2	0	0	0	90	150	Oxygen	115.28	6.1
3	1	1	0	120	200	Oxygen	95.76	6.4
4	0	−1	1	90	100	Argon	119.21	5.3
5	−1	0	−1	60	150	Argon	103.32	7.1
6	0	−1	−1	90	100	Argon	122.93	7.1
7	0	0	0	90	150	Oxygen	109.78	5.6
8	0	0	0	90	150	Oxygen	112.62	6.3
9	0	0	0	90	150	Oxygen	103.51	5.1
10	−1	0	−1	60	150	Argon	97.46	5.7
11	0	0	0	90	150	Oxygen	111.86	6.5
12	1	−1	0	120	100	Oxygen	107.83	7.0
13	−1	1	0	60	200	Oxygen	124.46	5.2
14	1	0	1	120	150	Argon	107.64	7.3
15	0	1	1	90	200	Argon	118.45	4.9
16	0	1	−1	90	200	Argon	111.34	6.3
17	1	0	−1	120	150	Argon	103.35	6.6

**Table 3 polymers-14-04232-t003:** ANOVA results of responses for plasma treated polyimide yarn.

Source	Extraction Force (cN)	Strain (%)
F	P		F	P	
Model	3.3	0.0647		2.00	0.1867	
A	0.0998	0.7613		10.39	0.0146	significant
B	0.4088	0.5429		1.45	0.2673	
C	0.7154	0.4256		5.96	0.0446	significant
AB	13.96	0.0073	significant	0.0593	0.8146	
AC	0.1350	0.7242		3.83	0.0913	
BC	0.7066	0.4284		0.1054	0.7550	
A^2^	7.94	0.0259	significant	0.0519	0.8263	
B^2^	0.8968	0.3752		0.0078	0.9323	
C^2^	1.38	0.2787		0.0220	0.8863	
Lack of Fit	5.09	0.0623		0.4289	0.6731

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
