# Peer review of "Surface Modification on Polyimide Yarn by Plasma Treatment to Enhance Adhesion with Polypropylene Resin"

_polymers, 2022, doi:10.3390/polym14194232_

Round 1
Reviewer 1 Report
Dear authors,
The manusript you submited is interesting, but requires major revision. My concerns and remarks are the following:
- Abstract: Some thing should be written more precisely, for example: what kind of plasma was used, what is the configuration of plasma generation? etc.
- Introduction:
Too many abbreviations which confuse the reader! Please omit some unncecessary abbreviations! Also the introduction is more like a literature review without any "connecting" text. Please improve!
L 42-43: ...the morphology of the fibers can be rough... - strange statement. Please improve.
L 46: "enhanced"- maybe better "elevated"?
L 58: TPS and SEM=?
L 63: DPF=?
L 65: Polypropylene fabric polyurethane coating = ? something is missing here
L 67: "in this study" - which study?
L 74: "surface colour strenght" = what?
L 82: "formation of nanoscale roughness" = ? please be more specific!
Section 2.1
Improve first sentence, it is not understandable.
L 121: "processing maschine for processing" = ?
What kind of plasma was used? Please details. Some sketch or figure is recommended.
Section 2.2 should be improved in a way that the reader at least imagine the procedure, how this is done... What is "extraction force" anyway?
L 155: "air" - did you mean "gas"?
Figure 1 and Figure 2 and others - The unit cN means what?
L 172: Optimal treatment time shoud be just one!
Table 3: "F" and "P" should be decribed. What means square (^2)?
L 196: "less" = lower?
L 201: "bad -- we" = ?
- The text is in general not understandable in some parts.
BR
Reviewer 2 Report
In this manuscript, the authors investigated the surface of polyimide yarn which was modified by plasma treatment. They studied the effect of plasma treatment parameters on the adhesion between polyimide yarn and polypropylene resin. They compared the extraction force on the surface of polyimide yarn before and after treatment with plasma. Furthermore, the adhesive force was analyzed by a single factor to optimize the process parameters to obtain a better adhesive force. The optimal process parameters are obtained through analysis: treatment time 90 s, the processing gas oxygen, and the output power 150 W. This study is good and important for the project of the plasma treatment technology which has a significant surface modification effect on polymers. It can effectively improve the adhesion between copolymers or/and composites. Also, the plasma processing environment is friendly, easy to operate, and has a good development prospect. This study can provide a reference for the further development and application of modified polyimide fiber yarns. The interpretations of the results are well discussed. The quantity and quality of the figures are appropriate. We believe that this research subject is promising for developing a high-performance fiber resin matrix composites.
Summary: I recommend publishing this manuscript after considering my comments on the attached file.

Reviewer 3 Report
In this article, using plasma treatment for modified polyimide yarn surface, and to study the plasma processing parameters influence on polyimide yarn and polypropylene resin adhesion, in addition, also use the plasma processing parameters were optimized Box - behnken design, discusses the plasma parameters on the influence of polyimide fiber and resin adhesive. Overall, the manuscript is well organized, clear, and the conclusions are supported by the experimental and results. However, there are still some issues that need to be addressed before its acceptance.
1. One sentence to present the background of this work should be added at the beginning of the abstract section.
2. In the introduction section, authors provided many examples on the plasma treatment on other fibers, which can be shortly introduced. Briefly introduction on the advantages of plasma treatment on fibers can be applied.
3. More details on the raw materials should be provided to ensure the repeat of this work.
4. The horizontal and vertical coordinates of Figure 1 and Figure 2 should be consistent, and the font size should be consistent.
5. When an acronym appears for the first time in an introduction, write the full name for easy understanding.
6. More introduction on the structure, properties and applications of polyimide fibers should be provided with supporting recent 3 years articles, such as: Electrospun polyimide nonwovens with enhanced mechanical and thermal properties by addition of trace plasticizer; Robust strong electrospun polyimide composite nanofibers from a ternary polyamic acid blend; High-density fibrous polyimide sponges with superior mechanical and thermal properties; Superior mechanical enhancement of epoxy composites reinforced by polyimide nanofibers via a vacuum-assisted hot-pressing; Mechanical and thermal properties of electrospun polyimide/rGO composite nanofibers via in-situ polymerization and in-situ thermal conversion; etc.
7. The reasons can be properly explained in the chart analysis of Figures 3 and 4.
8. Figure 6 coordinates are fuzzy, which is inconvenient for readers to understand.
9. Authors should pay attention to the writing of lower marks in references.
10. In results and discussion section, more discussion and comparison to previous reports should be provided.
11. Authors should carefully check the whole manuscript. There are still some typos and grammar issues.
Round 2
Reviewer 1 Report
Dear Authors,
Figures 1 and 2 are rather basic, but better than nothing...
Indications in Figure 2 must be checked (2 times resin bead? where is "fiber strength elongator" then?)
Some sentences are still long and also other mistakes in English grammar must be improved.
Best regards
Reviewer 3 Report
Accept in present form